# Assessment of a Bioimpedance-Based Method for the Diagnosis of Oral Cancer

**DOI:** 10.3390/diagnostics14242894

**Published:** 2024-12-23

**Authors:** Kristina Horvat Šikonja, Ivica Richter, Marko Velimir Grgić, Krešimir Gršić, Dinko Leović, Lovorka Batelja Vuletić, Vlaho Brailo

**Affiliations:** 1Department of Oral Medicine, School of Dental Medicine, University of Zagreb, 10000 Zagreb, Croatia; ivica.richter@ri.t-com.hr (I.R.); brailo@sfzg.unizg.hr (V.B.); 2Clinic for Otolaryngology and Head and Neck Surgery, Clinical Hospital Centre “Sestre Milosrdnice”, 10000 Zagreb, Croatia; marko_grgic@yahoo.com; 3Clinic for Otolaryngology and Head and Neck Surgery, University Clinical Hospital Centre Zagreb, 10000 Zagreb, Croatia; kresimir.grsic@gmail.com (K.G.); dinko.leovic@gmail.com (D.L.); 4Department of Pathology, School of Medicine, University of Zagreb, 10000 Zagreb, Croatia; lovorka.batelja.vuletic@mef.hr; 5Clinic for Dentistry, University Clinical Hospital Centre Zagreb, 10000 Zagreb, Croatia

**Keywords:** oral cancer, bioimpedance, oral mucosa, diagnostics

## Abstract

**Background/Objectives:** Oral cancer (OC) is a disease with poor prognosis mainly due to late diagnosis. There is considerable interest in the use and development of rapid, point of care (POC) non-invasive methods that can accelerate the diagnostic process. Bioimpedance (BI) is resistance to the passage of electric currents through tissue that reflects structural changes in the tissue. The aim of this study was to determine the spectrum of BI values in patients with oral cancer, to compare them with other oral lesions and healthy controls, and to determine the diagnostic value of the BI-based method for diagnosis of OC. **Methods:** Ninety-three participants divided into three groups participated in this study. The first group (31 participants) consisted of patients with histologically confirmed OC, the second group (31 participants) consisted of patients with an active reticular form of oral lichen planus (positive controls; OLP) and the third group (31 participants) consisted of healthy controls. In OC and OLP patients, BI was measured at three points (non-ulcerated lesional mucosa, clinically unaffected perilesional mucosa and unaffected mucosa on the contralateral side). In healthy controls, BI was measured on a healthy mucosa in the corresponding anatomical region. Measurements were performed at nine frequencies (1, 2, 5, 7, 10, 20, 70 and 100 kHz). **Results**: In OC patients, BI values in the lesion were significantly lower than BI values in clinically intact perilesional mucosa and the unaffected contralateral side at all frequencies. Furthermore, BI values of the clinically intact perilesional mucosa were significantly lower than the BI values of the healthy contralateral mucosa at frequencies of 1 kHz, 2 kHz, 5 kHz, 7 kHz and 10 kHz. Patients with OC had significantly lower BI values compared to patients with OLP and individuals with healthy oral mucosa at all frequencies. **Conclusions**: This study demonstrated the very good to excellent ability of this method to detect OC lesions, which needs to be confirmed by further studies on a larger number of participants.

## 1. Introduction

Oral cancer (OC), a disease which accounts for 3.5% of all cancer diagnoses, belongs to a group of diseases with high morbidity and mortality [1]. The largest share of OC is diagnosed in Asia (64.2%), followed by Europe (17.4%), North America (7.6%), South America (5.6%), Africa (3.8%) and Oceania (1.3%) [2,3]. Based on gender stratification, the incidence rate is significantly higher in men (13.1/100,000) than in women (5/100,000), and the risk of the disease increases with age [1].

The main risk factors for the development of OC are tobacco and alcohol use, with an exponential increase in risk in the presence of both factors. Approximately 70% of cancers in this region can be explained by exposure to one or both of the aforementioned risk factors [4]. The most common localizations of OC are the ventral and lateral tongue and the floor of the mouth (over 50% of cases), mostly in Western countries [5,6,7,8,9]. Other localizations are the buccal mucosa, retromolar area, maxillary and mandibular gingiva, soft palate and, less frequently, hard palate.

Pain is not a common symptom in patients with OC; it occurs in 30–40% of cases, and it usually appears when the lesions reach the advanced stage [10]. In later and larger lesions, symptoms can vary from mild discomfort to severe pain, especially in the tongue. Other symptoms include ear pain, bleeding, loose teeth, difficulty speaking, dysphagia, trismus, paraesthesia and difficulty using dentures [11].

Due to the advanced stage at the time of diagnosis, the disease has a poor prognosis, with an overall five-year survival rate of less than 55%. Treatment often includes surgical interventions, radiotherapy or chemotherapy, which significantly impairs the quality of life. Prognosis and survival rates undoubtedly depend on the stage of the lesion at the time of diagnosis and the rapid and adequate therapeutic response. Five-year survival rates drop significantly in patients with locally and regionally advanced disease, emphasizing the importance of early diagnosis [12].

Early diagnosis significantly improves prognosis and outcomes, reaching a 5-year survival rate of up to 90% [13]. Traditionally, the gold standard of screening for OC is a conventional examination of the oral cavity, and a definitive diagnosis is made on the basis of a lesion biopsy [5,14]. Biopsy, however, has several limitations. It is an invasive procedure that some patients may be reluctant to undertake. Next, it is usually performed in specialist institutions, and waiting for the procedure can cause additional stress for the patient. The biopsy should be performed by a specialist with experience in the management of oral mucosal lesions in order to avoid taking an unrepresentative sample. Because of its invasiveness, biopsy requires specific infection control precautions that require trained personnel and additional resources. Accordingly, there is considerable interest in the use and development of new, rapid, non-invasive methods for the assessment of lesions, which can be used at the point of care (POC) and which are not so dependent on subjective assessment, like, for example, an assessment of epithelial dysplasia [15,16,17,18]. The use of POC testing brings multiple benefits, both clinical and economic. Clinically, it enables faster diagnosis, immediate administration of therapy, better patient cooperation and a reduction in complications and administrative burdens, which lead to greater patient satisfaction. Economic benefits include a reduction in the number of clinical examinations, a shorter stay in health facilities, targeted interventions and a reduction in the loss of working days, which positively affect productive years and general economic indicators [15,16,17,18].

Bioimpedance (BI) is a measure that expresses resistance to the passage of electric current through tissue. BI is a characteristic of all living tissues and changes depending on the structure and chemical composition of the tissue [19]. Changes in structure and/or chemical composition result in changed electrical resistance of tissues. The application of BI for diagnostic purposes in medicine and dentistry is based on this principle. Due to their non-invasiveness, BI-based methods have certain advantages over invasive methods—they are simpler to perform, are more comfortable for the patient and present less of a problem in terms of disinfection, sterilization and infection control. The application of BI-based methods in medicine and dentistry is very diverse, from the examination of pathological changes in the tissue to the determination of the length of the root canal [20,21,22].

Due to the fact that different structural alterations in tissue facilitate the flow of electric current, i.e., lower tissue resistance, BI-based methods able to register these alterations are studied as a non-invasive method for the diagnosis of oral cancer (and other cancers as well). A recent systematic review identified five such studies [23]. All of the studies reported significantly lower BI values in OC compared to healthy tissue. The review concluded that due to its non-invasiveness, reliability and immediate results, BI appears to be a promising tool for oral cancer screening [24,25,26,27,28].

The aim of this study was to determine the spectrum of BI values in patients with oral cancer, to compare them with the values of other oral lesions and healthy subjects, and to determine the diagnostic value of the BI-based method for the diagnosis of OC.

## 2. Materials and Methods

This study was approved by the ethics committee of the University of Zagreb School of Dental Medicine (No. 05-PA-30-VIII-6/2019) and University Clinical Hospital Zagreb (No. 02/21 AG). Ninety-three participants divided into three groups participated in the study. Before the enrolment, all participants signed informed consent according to the Declaration of Helsinki.

The first group (31 participants) consisted of patients with histologically confirmed OC from the Department of Maxillofacial Surgery University Clinical Hospital Zagreb and University Clinical Hospital “Sestre Milosrdnice”. Patients with another type of cancer, patients with localization not suitable for the placement of the measuring electrode (oropharynx, gingiva and hard palate) and patients who were not able to comprehend the informed consent were excluded.

The second group (31 participants) consisted of patients from the University of Zagreb, School of Dental Medicine, Department of Oral Medicine with an active reticular form of oral lichen planus (positive controls; OLP). The diagnosis of OLP was established based on clinical and histological characteristics of the lesions. Exclusion criteria were similar to the ones applied in the OC group.

The third group (31 participants) consisted of patients from other departments of the University of Zagreb, School of Dental Medicine, with clinically normal oral mucosa, which was confirmed by an experienced oral medicine specialist (healthy controls; HC).

The intraoral sensor consisted of three concentric electrodes made of sintered aluminium alloys of high conductivity, coated with an insulating layer of Teflon, with a total diameter of 8 mm. In order to ensure stable contact and uniform pressure of the sensor on the oral mucosa, the sensor was connected to a dental suction unit which produced a constant negative pressure of 250 mBar, thus ensuring its stability during the measurement. The sensor was connected to the measuring device NI USB-6251 (National Instruments^®^, Austin, TX, USA) via electrical conductors, and the measuring device was connected to the laptop via the USB port. Measuring software, developed using the Lab View 8.5.1 software package (National Instruments^®^, Austin, TX, USA), converted electrical signals into digital records and stored them in a database. A schematic diagram of the measuring system is displayed in Figure 1. The measurement was simple and non-invasive—the electrode was placed on the selected place on the oral mucosa, and the system registered the measurement. After placing the sensor on the oral mucosa, an alternate current at 9 different frequencies passed between the electrodes through the tissue, and the voltage between the electrodes was measured. Measurements were performed at 9 frequencies (1, 2, 5, 7, 10, 20, 70 and 100 kHz). The device permits users to set the frequency range manually through the software from 0 to 100 kHz. Since there is no standard frequency range universally accepted for BI measurements of oral mucosa, the aforementioned frequencies were determined arbitrarily in order to assess which frequency yields the best discriminative ability to identify OC lesions.

In OC patients, BI was measured at 3 reference points:Morphologically changed but non-ulcerated mucosa of the OC lesion;Clinically intact mucosa in the immediate vicinity of the tumour, i.e., 5 mm away from the tumour border;Healthy oral mucosa of the corresponding anatomical region on the contralateral/unaffected side. Measurements on healthy contralateral mucosa were always performed in the same region as the tumour, but on the unaffected side.

The measurement was repeated in the same way in the group of positive controls, while in the group of negative controls BI was measured on a healthy mucosa in the corresponding anatomical region. Each measurement was repeated three times, and the mean value of the three measurements was used in further calculations.

Normality of distribution was assessed by the Shapiro–Wilk test. Since all variables (except age) deviated from a normal distribution, the median and interquartile range (IQR) were used to display BI values, while the mean value ± standard deviation was used to present age. Within-group differences in BI were assessed by Friedman’s analysis of variance of followed by the Dunn–Bonferroni post hoc test. BI differences between groups were assessed by the Kruskal–Wallis test followed by Dunn’s post hoc. The discriminative ability of the method for the identification of oral cancer was tested by receiver operating characteristic (ROC) curve analysis and classified according to the area under the curve (AUC) size as follows: >0.9, excellent; 0.8–0.9, very good; 0.7–0.8, good; 0.6–0.7, satisfactory. *p* values less than 0.05 (*p* < 0.05) were considered statistically significant.

## 3. Results

A total of 93 participants (53 women and 40 men) with an average age of 62.0 ± 11.5 years participated in this study. Demographic and clinical data on the participants are presented in Table 1. No statistically significant difference in age and gender between the participants was found (*p* = 0.713; *p* = 0.102). A statistically significant difference was found in the localization of the lesion/measurement point between the groups (*p* < 0.0001).

### 3.1. Bioimpedance Spectra in Patients with Oral Cancer

Figure 2 displays the results of BI measurements in the group of patients with OC. The BI values of the lesion were significantly lower than the BI values of the clinically intact perilesional mucosa and the BI values of healthy mucosa contralaterally at all frequencies (*p* < 0.05). Furthermore, the BI values of the clinically intact perilesional mucosa were significantly lower than the BI values of the healthy contralateral mucosa at frequencies of 1 kHz, 2 kHz, 5 kHz, 7 kHz and 10 kHz (*p* < 0.05) (Figure 2).

### 3.2. Comparison of Bioimpedance Values in the Lesion Between Oral Cancer Patients and Controls

Figure 3 displays a comparison of BI values in the lesion between the three groups of participants. Patients with OC had significantly lower BI values in the lesional tissue compared to lesional tissue in positive controls and healthy controls at all measured frequencies (*p* < 0.05), except at the frequency of 100 Hz. At the frequency of 100 Hz, BI values in the OC lesion were significantly lower than BI values of the lesional tissue in positive controls (*p* < 0.0001) and lower but not significantly different than BI values in healthy controls (*p* = 0.064) (Figure 3).

### 3.3. Comparison of Bioimpedance Values in the Clinically Intact Perilesional Mucosa Between Oral Cancer Patients and Controls

Figure 4 displays a comparison of BI values on the clinically intact perilesional mucosa between three groups of participants. The BI values of the clinically intact perilesional mucosa in OC patients were lower than the BI values of the clinically intact perilesional mucosa in positive controls and the BI values in healthy controls at the frequencies of 1 kHz, 2 kHz, 5 kHz, 7 kHz and 10 kHz (*p* < 0.05). At higher frequencies (20 Hz, 50 Hz, 70 Hz and 100 Hz), the BI values of the clinically intact perilesional mucosa in OC patients were significantly lower than the BI values of the clinically intact perilesional mucosa in positive controls but not significantly different from the BI values in healthy controls (Figure 4).

### 3.4. Assessment of the Ability of the Bioimpedance-Based Method to Identify Oral Cancer Lesions

Assessment of the ability of the BI-based method to identify oral cancer lesions was performed by ROC curve analysis. The analysis showed the excellent to very good discriminating ability of the method, with AUC values from 0.958 to 0.792, depending on the measurement frequency. The AUC decreased with the increase in measurement frequency (Figure 5).

## 4. Discussion

The results of this study demonstrated that OC lesions have lower BI values compared to the surrounding healthy tissue. The EI values of the OC and surrounding healthy tissue decreased as the frequency of measurement increased, and the differences with regard to the type of tissue were less pronounced. Lower BI values in OC lesions compared to healthy tissue are a result of changes in the electrical properties of cancer tissue. At low frequencies, electric current moves through intercellular spaces without the possibility of penetrating cells, and the BI values are dominated by the results from the most superficial layer of the oral mucosa [20]. In healthy tissue, the cells are densely packed and tightly connected to each other. The intercellular space is very narrow and provides higher resistance to the low-frequency current. On the other hand, tumour tissue has a larger extracellular space due to the loss of intercellular connections, more extracellular matrix, the altered permeability of the cell membrane, the reduced cell density and the different cell orientation, resulting in a drop in resistance and easier penetration of current into the intercellular space [21,22]. Similar results were reported in the studies of Sun et al. [24], Ching et al. [25], Sarode et al. [26] and Murdoch et al. [27]. The authors explained the obtained differences in EI values between tumour tissue and healthy mucosa with ultrastructural changes in the tumour tissue and the consequent changes in the electrical properties of tissues and cells described earlier. This is further supported by the study by Sarode et al., who reported statistically significant differences between well- and poorly differentiated carcinomas of the oral cavity [26]. It is known that poorly differentiated cancers have a more altered tissue architecture and more intercellular space compared to well-differentiated cancers, which is why the resistance to the flow of electric current is lower. In this study, it was not possible to determine the difference between poorly and well-differentiated cancers, because the number of well-differentiated cancers was several times greater than that of poorly differentiated cancers (27 vs. 4) among our patients.

Regarding the differences between the intact perilesional mucosa and the healthy contralateral mucosa, the EI values at frequencies from 1 kHz to 10 kHz were significantly lower than the EI values on the healthy contralateral mucosa. At higher frequencies (20–100 kHz), the impedance of the intact perilesional mucosa was lower than that of the contralateral healthy mucosa, but not enough to achieve a statistically significant difference. Perilesional tissue, up to 1 cm from the edge of the malignant lesion, although it may appear macroscopically healthy, may show changes compared to normal oral tissue [29]. EI values of perilesional tissue may be higher than in cancer lesions, but lower than in completely healthy tissue due to the presence of subclinical changes or initial stages of malignant tissue transformation that are not yet clinically visible. The obtained results could not be compared with the results of other studies, because there were no studies in the available literature that compared EI values on perilesional clinically unchanged mucosa and healthy mucosa in patients with OC.

A comparison between the groups revealed that the EI values measured on OC lesions were significantly lower compared to the EI values of healthy controls at all frequencies, except for the frequency of 100 kHz, where the values were lower, but the difference was not statistically significant. Furthermore, the values of EI in OC lesions were significantly lower at all frequencies than positive controls, i.e., of EI values on lesions in patients with OLP. This finding is not surprising because patients with OLP, despite subepithelial inflammation and degeneration of basal cells, have preserved epithelial stratification and proper cell orientation, which is not the case with the cancer tissue. In addition, patients with OLP have hyperkeratosis as an additional factor that negatively affects electrical conductivity even in healthy tissue. A study by Richter et al. reported the highest EI values in healthy participants on the hard palate, a region that has the thickest corneal layer in the entire oral cavity [30].

Regarding the comparison of EI values on perilesional mucosa, it was found that EI values on clinically intact perilesional mucosa in patients with OC were lower than EI values on clinically unaffected mucosa of patients with OLP and healthy controls at lower frequencies (1 kHz, 2 kHz, 5 kHz, 7 kHz and 10 kHz). At higher frequencies (20 kHz, 50 kHz, 70 kHz and 100 kHz), significant differences were found only between clinically intact perilesional mucosa in patients with OC and patients with OLP. The results cannot be compared with the results of other studies because to our knowledge no studies in the available literature measured EI on the clinically intact perilesional mucosa in OC patients. This result can be explained by the fact that the low-frequency current passes through the intercellular space in contrast to the high-frequency current that passes through the cells [22]. Measuring EI values at low frequencies could possibly detect more subtle changes in the tissue that are not yet clinically visible. This needs to be confirmed by further studies, and if found to be correct, could represent a potential method for preoperative assessment of tumour margins. The distance of the tumour from the resection margin is a factor known to significantly influence local disease control and patient survival [31,32,33].

The results showed a good to excellent ability of the method to identify OC lesions (AUC 0.792–0.958). AUC values were highest at the frequency of 1 Hz and gradually decreased with the increase in frequency. This is not surprising considering the previously mentioned property of the lower-frequency electric current that passes through the intercellular spaces and thus better reflects changes in the tissue structure, in contrast to the higher-frequency current that passes through the cells [22]. In patients with OC, however, changes also exist at the cellular level, which is why measurements at higher frequencies were able to identify OC lesions as well. However, the best diagnostic performance (sensitivity of 96.8% and specificity of 87.1%) was achieved at the lowest frequency, i.e., 1 Hz.

The above results are in agreement with studies of other authors that also reported the very good ability of EI-based methods to identify OSCC. In an in vitro study by Carobbio et al., EI values were determined on samples of different tissues affected by OSCC at frequencies of 10–100 kHz [34]. On 384 mucosal samples, the device achieved very good discriminatory ability with an AUC value of 0.81, a sensitivity of 87% and a specificity of 76%. In the study by Murdoch et al. [27], which tested a commercially available device (ZedScan^®^, Zilico Ltd., Manchester, UK) intended to aid in the diagnosis of cervical lesions, the AUC value was 0.776, sensitivity 65.2% and specificity 91%. The device used a frequency range of 0.076–625 kHz, and 10 subjects with OC, 37 positive controls (patients with dysplastic lesions of varying degrees) and 51 healthy subjects participated in the study. Tatullo et al. reported the very good ability of an EI-based device to detect OLP lesions by measuring EI at a frequency of 50 kHz in 52 OLP patients and 11 control subjects [28]. The AUC value in the mentioned study was 0.89. The differences between the results of this study and the results in the literature can be explained by the different constructions of the devices, different measurement frequencies and different numbers of participants.

This study has several limitations that need to be mentioned. It was conducted on a relatively small number of respondents (31), and based on that number it is not possible to draw definitive conclusions about the effectiveness of the method/device. The number of participants in our study was not significantly different from the number of participants in the other studies—5 participants with OC in the study by Ching et al., 12 participants with OC in the study by Sun et al., and 10 participants with OC and 37 subjects with different dysplastic lesions of the oral mucosa in the study by Murdoch et al. [24,25,27]. A study by Sarode et.al. involved 50 patients with OC, and the study by Tatullo et al. involved 52 participants with OLP [26,28]. Regardless of the number of participants, the trend of decreased BI values in OC patients is obvious and needs to be further evaluated in larger studies.

One might note a slight male predominance in the OC group and a slight female predominance in the OLP group, which was a consequence of the disease epidemiology (i.e., higher prevalence of OC in males compared to females and vice versa for OLP). This predominance, however, did not reach statistical significance. Even though females are reported to have higher BI values of oral mucosa compared to males [30,35], we believe that the difference between the groups was not caused by different prevalences of males and females in both groups but by structural changes in tissue architecture caused by tumour formation. This is supported by significant differences in the BI values of the OC lesions and contralateral healthy mucosa in OC patients, as BI values are more affected by local changes in tissue than factors such as age and sex.

As for the selection of participants, patients with OC all had histologically confirmed disease that also had a clear clinical presentation, for which an experienced clinician did not need any auxiliary diagnostic tool. However, the purpose of this study was to evaluate a method that can differentiate a suspicious lesion well, and it was necessary to test it precisely on histologically confirmed OC lesions. A diagnostic device based on this method could, at the level of a general dentist, accelerate the decision to refer patients to specialist treatment and potentially shorten the “second lost time” in the process of OC diagnosis [36]. In specialist practice, such a device could speed up the decision on biopsy of a suspicious lesion. As for the control group, patients with active reticular OLP were selected as a positive control because the lesions of active reticular OLP have a marked red and white component, elements that are also visible in erythroleukoplakia, a lesion that most often represents the early stage of OC [37,38]. We wanted to have a group of positive controls that would resemble the early stage of OC as much as possible. Therefore, patients with ulcerations of other aetiologies were not included as positive controls, as ulceration represents a more advanced stage of OC. This was also not the case in other studies on BI and oral cancer [23,24,25,26,27,28]. There are no data in the literature about the BI values of oral ulcerations of other aetiologies, so one can only speculate about the differences between such lesions and OC. However, it is unlikely that the inflammatory infiltrate underlying such lesions would affect the electrical resistance of the tissue in the same way as the altered tissue architecture present in OC lesions does.

## 5. Conclusions

In conclusion, the results of this study demonstrated that OC lesions have a lower spectrum of BI values compared to OLP lesions and healthy oral mucosa. The study demonstrated the very good to excellent ability of this method to detect OC lesions, which needs to be confirmed by further studies on a larger number of participants.

## Figures and Tables

**Figure 1 diagnostics-14-02894-f001:**
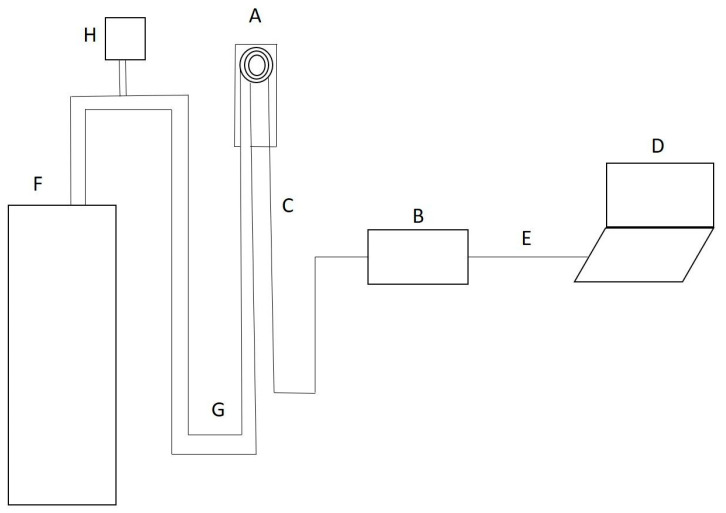
Schematic diagram of the measuring system. The intraoral sensor (A) was connected to the measuring device (B) with an electric conductor (C). The measuring device (B) registered resistance to the flow of the electric current. The device (B) was connected to the laptop (D) by a USB cable (E). Analytical software stored on the laptop (D) converted the electrical signal from the device (B) into digital data. To maintain the constant pressure of the sensor (A) on the oral mucosa, the intraoral sensor was connected to the dental suction unit (F) by a rubber tube (G). A constant pressure of 250 mBar was maintained and controlled with a manometer (H).

**Figure 2 diagnostics-14-02894-f002:**
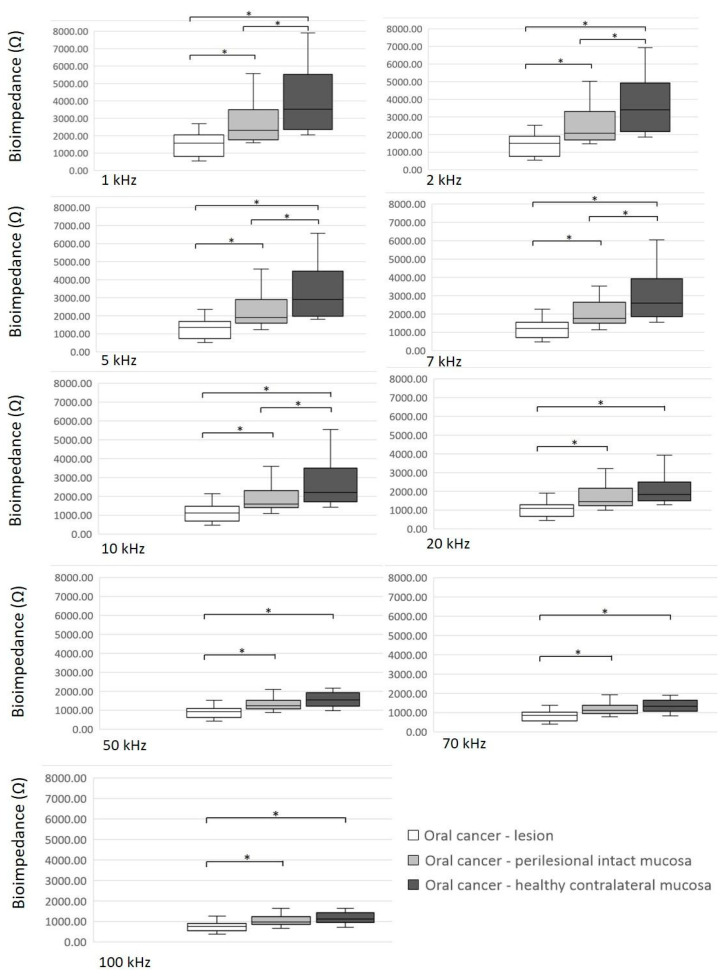
Bioimpedance measurements in a group of patients with oral cancer. Significant differences (*p* < 0.05) are marked with an asterisk (*).

**Figure 3 diagnostics-14-02894-f003:**
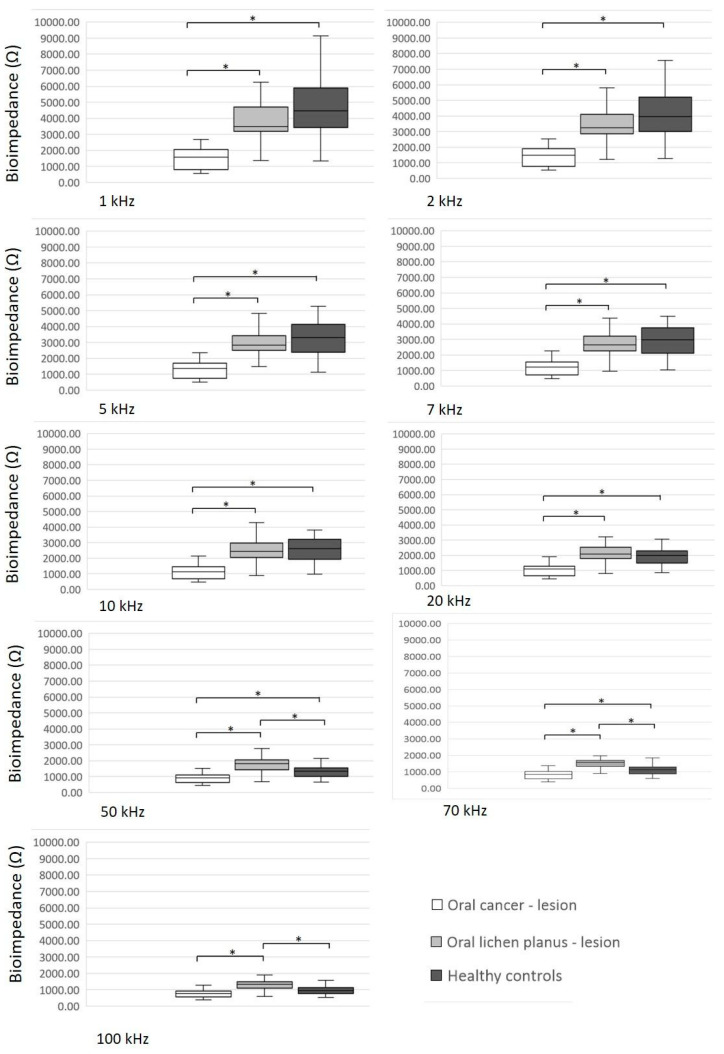
Comparison of bioimpedance values of the lesion between three groups of participants. Significant differences (*p* < 0.05) are marked with an asterisk (*).

**Figure 4 diagnostics-14-02894-f004:**
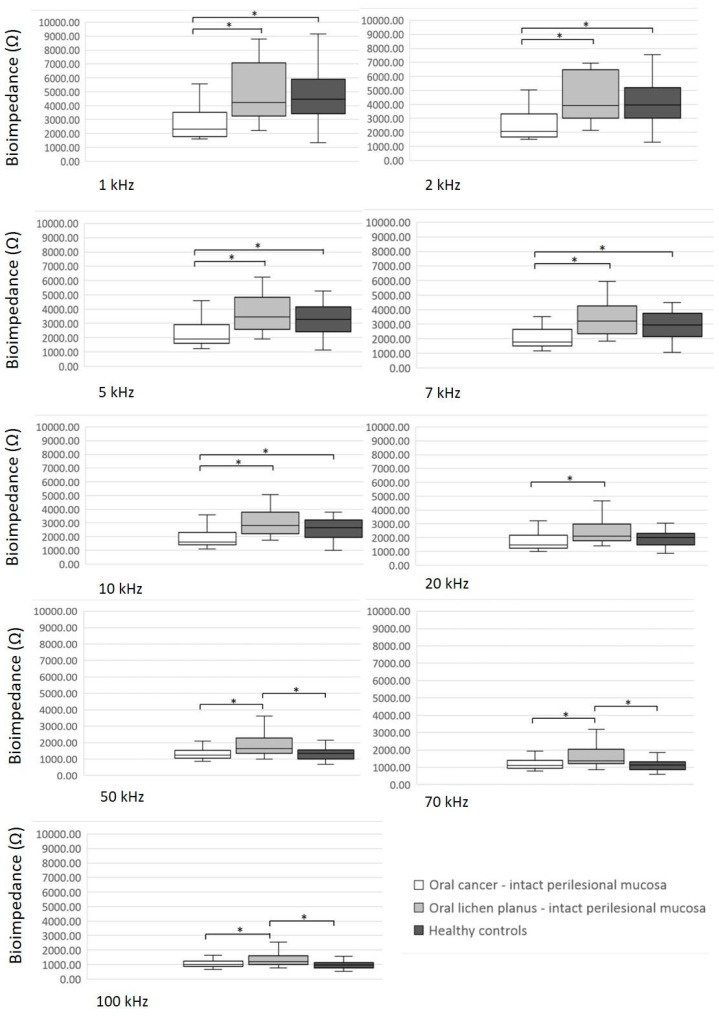
Comparison of bioimpedance values of the clinically intact perilesional mucosa between three groups of participants. Significant differences (*p* < 0.05) are marked with an asterisk (*).

**Figure 5 diagnostics-14-02894-f005:**
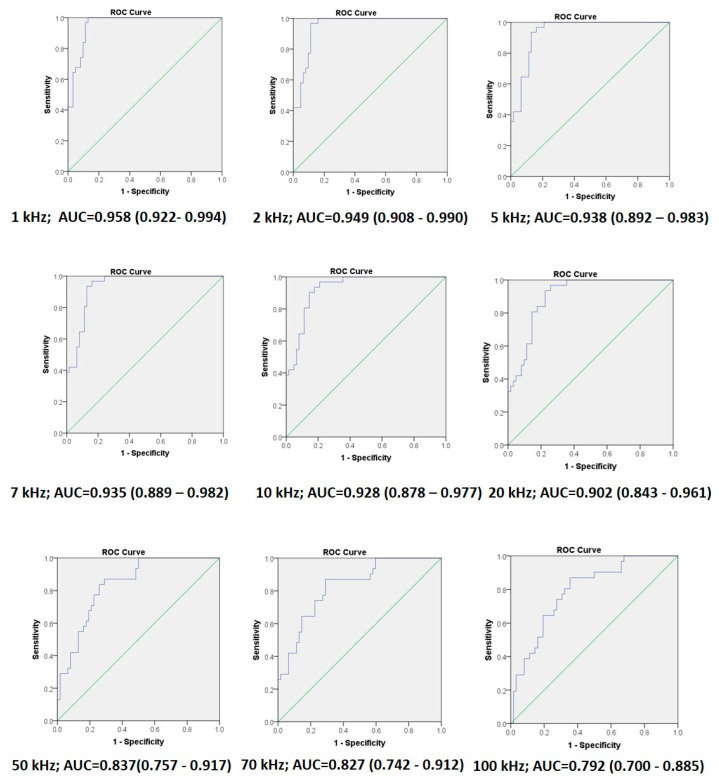
Analysis of the discriminatory ability of the method for the detection of oral cancer on frequencies from 1 kHz to 100 kHz. The area under the curve (AUC) and 95% confidence intervals (95% CI) at corresponding frequencies are displayed below each graph. The AUC was highest at the frequency of 1 kHz and decreased as the frequency increased.

**Table 1 diagnostics-14-02894-t001:** Demographic and clinical characteristics of the participants.

	Oral Cancer	Positive Controls	Healthy Controls	*p*
Sex, N (%)				
Male	21 (67.7)	13 (41.9)	19 (61.3)	0.102
Female	10 (31.3)	18 (58.1)	12 (38.7)	
Age (mean ± SD)	62.1 ± 12.8	63.2 ± 11.2	60.8 ± 10.7	0.713
Localization of the lesion, N (%)				
Buccal mucosa	10 (32.3)	28 (90.3)	Not applicable	*p* < 0.0001 *
Tongue border	12 (38.7)	3 (9.7)		
Floor of the mouth	9 (29)	0		
TNM stage				
1	4	Not applicable	Not applicable	Not applicable
2	4			
3	11			
4	12			
Tumour differentiation, N (%)				
Well differentiated	27 (87.1)	Not applicable	Not applicable	Not applicable
Poorly differentiated	4 (12.9)			

* *p* < 0.05.

## Data Availability

The data presented in this study are available on request from the corresponding author.

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
