# Peer review of "Assessment of a Bioimpedance-Based Method for the Diagnosis of Oral Cancer"

_diagnostics, 2024, doi:10.3390/diagnostics14242894_

Round 1
Reviewer 1 Report
Comments and Suggestions for Authors
1. What does Localization of the lesion mean for the healthy control in Table 1? Should it say - not applicable?
2. Line 144 - 62 ± 11.5 - is incorrect, it should be 62.0 ± 11.5 or 62 ± 12.
3. I would like to see a little more information about the method for determining bioimpedance in the Materials and Methods section, since the method is not widely used and is not familiar to readers. Can you provide a diagram of the device.
4. After treatment for lichen planus, do bioimpedance values change in patients? Can this indicator be used for monitoring during therapy?
5. You need to write in more detail about the different frequencies of bioimpedance, why these frequencies were used, is it possible to analyze only one of these frequencies? Do they have different clinical significance?
6. The study included patients with different stages of oral cancer and different degrees of tumor cell differentiation. Are there any differences between the bioimpedance values in these subgroups? Are there any gender differences? Should age be taken into account when comparing with healthy controls?
Author Response
Reviewer 1
Dear reviewer, many thanks for your valuable input that significantly helped us to improve the manuscript.
- What does Localization of the lesion mean for the healthy control in Table 1? Should it say - not applicable?
Many thanks for Your kind remark. This has been corrected in the table 1.
- Line 144 - 62 ± 11.5 - is incorrect, it should be 62.0 ± 11.5 or 62 ± 12.
Many thanks for pointing this out. This has been changed to 62.0 ± 11.5.
- I would like to see a little more information about the method for determining bioimpedance in the Materials and Methods section, since the method is not widely used and is not familiar to readers. Can you provide a diagram of the device.
We have added the following section in the methods for clarification:
“After placing the sensor on oral mucosa, alternate current at 9 different frequencies passed between the electrodes through the tissue and the voltage between them was measured. Measurements were performed at 9 frequencies (1, 2, 5, 7, 10, 20, 70 and 100 kHz). The device permits users to set the frequency range manually through the software from 0-100kHz. Since there is no standard frequency range universally accepted for BI measurements of oral mucosa, the aforementioned frequencies were determined arbitrarily in order to assess which frequency yields the best discriminative ability to identify OC lesions. “
The diagram of the device has been included in the manuscript as well (Figure 1).
- After treatment for lichen planus, do bioimpedance values change in patients? Can this indicator be used for monitoring during therapy?
We did not measure BI values pre and post treatment in OLP patients as this was a cross-sectional study. All OLP patients were in active reticular form, since we wanted the lesion to have marked red and white component to resemble erithroleukoplakia i.e. early stage of oral cancer.
To our knowledge no study assessed BI values pre and post treatment in patients with OLP. Based on the studies from the literature, we can (only) assume that values in OLP patients would change post treatment. Tatullo et al. (28. Tatullo M, Marrelli M, Amantea M, Paduano F, Santacroce L, Gentile S, et al. Bioimpedance Detection of Oral Lichen Planus Used as Preneoplastic Model. J Cancer. 2015;6(10):976–83.) reported significantly lower BI values in patients with erosive form of OLP compared to healthy individuals and individuals with reticular forms of OLP. However, we need to emphasize that in this study measurements were done in only 3 patients with erosive form of OLP. The inflammation itself affects BI by facilitating the flow of electric current due to increased volume of extracellular fluid, tissue oedema, and vasodilatation. This notion is supported by study of Nicader et al. (Nicader I, Ollmar S. Electrical impedance related to structural differences in the skin and in the oral mucosa. Med Biol Eng Comput. 1999;37:161-2.) who reported significantly lower BI values in oral mucosa treated with chemical irritants compared to healthy contralateral mucosa. Similar findings were reported in the skin as well (Nicader I. Electric impedance related to exprimentally induced changes of human skin and oral mucosa. Stockholm [dissertation]. Kongl Carolinska Medico Chirurgiska Institutet; 1998.; Ollmar S, Eek A, Sundström F, Emtestam L. Electrical impedance for estimation of irritation in oral mucosa and skin. Med Prog Technol. 1995;2:29-37.).
Therefore we believe that EI values could change in response to OLP therapy, but it needs to be confirmed by further studies.
- You need to write in more detail about the different frequencies of bioimpedance, why these frequencies were used, is it possible to analyze only one of these frequencies? Do they have different clinical significance?
Thank You for pointing out our omission to explain the frequencies of the measurements. The device measures bioimpedance at 9 selected frequencies from 0-100 kHz. Each frequency can be set manually by the software in 1 Hz increments. Since there are no universally accepted frequency ranges, and there is no consensus on the “optimal frequency we selected frequencies of 1, 2, 5, 7, 10, 20, 70 and 100 kHz arbitrarily as we wanted to assess which frequency provides best discriminatory ability to identify oral cancer lesions. Theoretically, measurements could have been done on just one frequency (by setting values of other 8 frequencies to 0), but as we said, we wanted to assess the frequency that offers best diagnostic properties.
We added a section in the Methods for further clarification: “Measurements were performed at 9 different frequencies (1, 2, 5, 7, 10, 20, 70 and 100 kHz). The device permits user to set the frequency range manually through the software from 0-100kHz. The aforementioned frequencies were determined arbitrarily in order to assess which frequency yields the best discriminative ability to identify OC lesion.”
We also added a sentence in the discussion: “AUC values were highest at the frequency of Hz and gradually decreased with the increase in frequency.” And “However, the best diagnostic performance (sensitivity 96.8% and specificity 81%) was achieved at the lowest frequency i.e. 1 Hz.”.
- The study included patients with different stages of oral cancer and different degrees of tumor cell differentiation. Are there any differences between the bioimpedance values in these subgroups? Are there any gender differences? Should age be taken into account when comparing with healthy controls?
Regarding the stage and differentiation, we did not perform a subgroup analysis due to small number of patients in each group (Stage 1 - 4 patients, Stage 2 - 4 patients, Stage 3 -11 patients and Stage 4 -12 patients; Well differentiated 27 patients vs. Poorly differentiated 4 patients). Making comparison with so little groups increases the chance of type I error. This is however, worth of further investigation on larger series of patients.
Age and gender are factors that can certainly affect BI values. In our previous studies (Richter I, Alajbeg I, Vučićević Boras V, Andabak Rogulj A, Brailo V. Mapping Electrical Impedance Spectra of the Healthy Oral Mucosa: a Pilot Study. Acta Stomatol Croat. 2015;49(4):331–9. and Horvat K, Richter I, Vucelić V, Gršić K, Leović D, Škrinjar I, Andabak Rogulj A, Velimir Grgić M, Brailo V. Impact of Age and Sex on Electrical Impedance Values in Healthy Oral Mucosa. Bioengineering (Basel). 2022 Oct 21;9(10):592. doi: 10.3390/bioengineering9100592.) we reported significantly higher BI values in women compared to men, and in younger people (20-39 yrs) compared to older (40-59 and 60+yrs, respectively). This is why we selected control groups to match our OC patients by sex and age. This is confirmed by the lack of significant differences in sex and age between groups, even though there was a slight male predominance in OC group and female predominance in OLP group. We believe that the aforementioned predominance is a consequence of the disease epidemiology (i.e. higher prevalence of oral cancer in males compared to females and vice versa for OLP) and that it did not affect BI values. We believe that BI values were mainly affected by structural changes in tissue architecture caused by tumor formation which is supported by significant differences in OC lesion and contralateral healthy mucosa in OC patients.
To further clarify this, we added a sentence in the discussion:
“One might note slight male predominance in the OC group and a slight female predominance in the OLP group which was a consequence of the disease epidemiology (i.e. higher prevalence of OC in males compared to females and vice versa for OLP). This predominance however did not reach statistical significance. Even though females are reported to have higher BI values of oral mucosa compared to males (30,35), we believe that the difference between the groups was not caused by different prevalence of males and females in both groups but by structural changes in tissue architecture caused by tumor formation. This is supported by significant differences in BI values of OC lesion and contralateral healthy mucosa in OC patients as BI values are more affected by local changes in tissue than factors such as age and sex.”
Reviewer 2 Report
Comments and Suggestions for Authors
This present manuscript is interesting, particularly in its exploration of using bioimpedance (BI) for oral cancer diagnosis. The manuscript is well-prepared but requires several clarifications and responses to enhance its clarity and scientific rigor:
1. Bioimpedance (BI) is a method used to estimate the fluid volume in tissues. However, there is currently no evidence to support the presence of fluid in oral cancer tissues. It would be beneficial to reference previous studies that have measured bioimpedance in oral cancer to strengthen the rationale for using this method.
2. Provide a more detailed explanation of how BI was measured in this study, including the range of frequencies used. Additionally, clarify how the BI values (omega values) were calculated.
3. The authors selected cancer localizations in the buccal mucosa, tongue, and floor of the mouth. It would be valuable to present the BI values for each anatomical region separately. This could help identify specific BI patterns associated with different cancer locations. Consider presenting this data in Figure 1.
4. Regarding Figure 1, were the BI measurements performed at consistent anatomical sites? For example, if the cancer is on the tongue, was the contralateral healthy mucosa measured at the buccal mucosa or floor of the mouth? Similarly, if the cancer is in the buccal mucosa, was the contralateral site measured on the tongue or floor of the mouth? If measurements were taken across different anatomical sites, it could introduce data bias.
5. Concerning the OLP diagnosis, what specific clinical types of oral lichen planus (OLP) were included in this study? For instance, erosive lichen planus is often considered an oral potentially malignant disorder (OPMD) and may exhibit malignant characteristics.
6. The conclusion states, “The study demonstrated very good to excellent ability of the method to detect OC lesions.” At which frequency or frequency range was this performance achieved?
Author Response
Reviewer 2
This present manuscript is interesting, particularly in its exploration of using bioimpedance (BI) for oral cancer diagnosis. The manuscript is well-prepared but requires several clarifications and responses to enhance its clarity and scientific rigor:
Dear reviewer, many thanks for your valuable input that significantly helped us to improve the manuscript.
- Bioimpedance (BI) is a method used to estimate the fluid volume in tissues. However, there is currently no evidence to support the presence of fluid in oral cancer tissues. It would be beneficial to reference previous studies that have measured bioimpedance in oral cancer to strengthen the rationale for using this method.
We agree that there is no evidence to support the presence of fluid in oral cancer. We changed the wording from “extracellular fluid” to “extracellular matrix”.
The use of BI for the diagnosis of oral cancer is based on morphological alterations in cancerous tissue that can facilitate the flow of electric current like increased amount of extracellular matrix, irregular stratification and orientation of cells, increased intercellular space etc.
In order to clarify this a section has been added to the introduction:
“Due to the fact that different structural alterations in tissue facilitate the flow of electric current i.e. lower tissue resistance, BI based methods able to register these alterations are studied as a non-invasive method for the diagnosis of oral cancer (and other cancers as well). Recent systematic review identified 5 such studies (24-28). All of the studies reported significantly lower BI values in OC compared to healthy tissue. The review concluded that due to its non-invasiveness, reliability and immediate results BI appears to be a promising tool for oral cancer screening. “
- Provide a more detailed explanation of how BI was measured in this study, including the range of frequencies used. Additionally, clarify how the BI values (omega values) were calculated.
The device measures bioimpedance at 9 selected frequencies from 0-100 kHz. Each frequency can be set manually by the software in 1 Hz increments. Since there are no universally accepted frequency ranges, and there is no consensus on the “optimal frequency" we selected frequencies of 1, 2, 5, 7, 10, 20, 70 and 100 kHz arbitrarily as we wanted to assess which frequency provides best discriminatory ability to identify oral cancer lesions. Theoretically, measurements could have been done on just one frequency (by setting values of other 8 frequencies to 0), but as we said, we wanted to assess the frequency that offers best diagnostic properties.
We added a paragraph for further clarification: “After placing the sensor on oral mucosa, alternate current at 9 different frequencies passed between the electrodes through the tissue and the voltage between them was measured. Measurements were performed at 9 frequencies (1, 2, 5, 7, 10, 20, 70 and 100 kHz). The device permits users to set the frequency range manually through the software from 0-100kHz. Since there is no standard frequency range universally accepted for BI measurements of oral mucosa, the aforementioned frequencies were determined arbitrarily in order to assess which frequency yields the best discriminative ability to identify OC lesions.”
- The authors selected cancer localizations in the buccal mucosa, tongue, and floor of the mouth. It would be valuable to present the BI values for each anatomical region separately. This could help identify specific BI patterns associated with different cancer locations. Consider presenting this data in Figure 1.
Thank you for your valuable suggestion. We opted for presenting the results from all 3 localizations summarize in one figure due to the following: a) In each subgroup the number of participants was small (10,12, 9) which could increase the chance of type 1 error; b) We performed a subgroup analysis and found no significant difference in BI values of OC lesions from different localizations c) According to the results of our earlier study (Richter I, Alajbeg I, Vučićević Boras V, Andabak Rogulj A, Brailo V. Mapping Electrical Impedance Spectra of the Healthy Oral Mucosa: a Pilot Study. Acta Stomatol Croat. 2015;49(4):331–9.) no significant difference in BI values between different sites in healthy individuals was found (apart from the hard palate, a site which was not affected in our patients).
We hope that You would accept our arguments for presenting the data this way.
- Regarding Figure 1, were the BI measurements performed at consistent anatomical sites? For example, if the cancer is on the tongue, was the contralateral healthy mucosa measured at the buccal mucosa or floor of the mouth? Similarly, if the cancer is in the buccal mucosa, was the contralateral site measured on the tongue or floor of the mouth? If measurements were taken across different anatomical sites, it could introduce data bias.
In oral cancer patients BI measurements on healthy contralateral mucosa were performed in the same region as the tumor, but on the unaffected site i.e. if the tumor was localized on the left lateral tongue, healthy contralateral mucosa was measured on the right lateral tongue, if the tumor was localized on the left buccal mucosa, healthy contralateral mucosa was measured on the right buccal mucosa, and if the tumor was localized on the left side of the floor of the mouth, healthy contralateral mucosa was measured on the right side of the floor of the mouth.
We added a sentence in the methods to further clarify this: “Measurements on healthy contralateral mucosa were always performed in the same region as the tumor, but on the unaffected side.”
- Concerning the OLP diagnosis, what specific clinical types of oral lichen planus (OLP) were included in this study? For instance, erosive lichen planus is often considered an oral potentially malignant disorder (OPMD) and may exhibit malignant characteristics.
In our study only patients with active reticular form of OLP were included. We specifically selected these patients due to the fact that these patients have marked red and white component of the lesion that could resemble erythroleukoplakia, the earliest stage of oral cancer. Even though malignant transformation is more common in erosive form of OLP, all forms of OLP can progress to oral cancer. Patients with erosive form were not included because erosive form of OLP is clinically quite distinct from erythroleukoplakia, and we wanted to have lesion that clinically resembles leukoplakia.
A section in the Discussion further clarifies this:
“As for the control group, patients with active reticular OLP were selected as a positive control because the lesions of active reticular OLP have a marked red and white component, elements that are also visible in erythroleukoplakia, a lesion that most often represents the early stage of OC [35,36]. We wanted to have a group of positive controls that would resemble the early stage of OC as much as possible. Therefore, patients with ulcerations of other aetiology were not included as positive controls, as ulceration represents more advanced stage of OC.”
- The conclusion states, “The study demonstrated very good to excellent ability of the method to detect OC lesions.” At which frequency or frequency range was this performance achieved?
The AUC i.e. the diagnostic ability of the method to identify OC lesions was highest at the frequency of 1Hz and it decreased with increasing frequency. In figure 3. AUC with 95% confidence interval is displayed at corresponding frequency below each graph. We edited the figure in order to make AUC values more visible. We also added a further clarification in the figure legend: “Area under the curve and 95% confidence intervals AUC (95%CI) at corresponding frequencies are displayed below each graph”.
Round 2
Reviewer 1 Report
Comments and Suggestions for Authors
I have no more comments on the manuscript.
Author Response
I have no more comments on the manuscript
Dear reviewer, we are thankful that You found the revised version of the manuscript suitable for publication. We really appreciate Your comments that helped us improve the manuscript.
Reviewer 2 Report
Comments and Suggestions for Authors
Author can improve the result section in abstract, by explain detail what are the result and what is the finding, not only describe in one sentences. as well as the methods, author can explain that BI was used with several frequency.
In introduction section, be careful during making paragraphs. one paragraphs at least 2 sentences.
Author Response
Author can improve the result section in abstract, by explain detail what are the result and what is the finding, not only describe in one sentence. As well as the methods, author can explain that BI was used with several frequency.
Many thanks for your remark. We have expanded the methods and results in the abstract according to Your instructions with the following section:
In OC and OLP patients BI was measured at 3 points (non-ulcerated lesional mucosa, clinically unaffected perilesional mucosa and unaffected mucosa on the contralateral side) In healthy controls BI was measured on a healthy mucosa in the corresponding anatomical region. Measurements were performed at 9 frequencies (1, 2, 5, 7, 10, 20, 70 and 100 kHz) Results: In OC patents BI values in the lesion were significantly lower than BI values in clinically intact perilesional mucosa and unaffected contralateral side at all frequencies. Furthermore, BI values of the clinically intact perilesional mucosa were significantly lower than the BI values of the healthy contralateral mucosa at frequencies of 1 kHz, 2 kHz, 5 kHz, 7 kHz and 10 kHz.
In introduction section, be careful during making paragraphs. one paragraphs at least 2 sentences.
You are absolutely right; it looks kind of weird. We have merged several sections into one larger paragraph. Many thanks for pointing that out.